REGISTERED REPORT PROTOCOL

# Effect of attention on human direction-discrimination thresholds at iso-eccentric locations in the visual field: A registered report protocol

Pankhuri Saxena[1,2]*, Stefan Treue[1,3,4,5]

**1** Cognitive Neuroscience Laboratory, German Primate Center-Leibniz Institute for Primate Research, Göttingen, Germany, **2** Göttingen Graduate Center for Neurosciences, Biophysics, and Molecular Biosciences (GGNB), University of Göttingen, Göttingen, Germany, **3** Faculty of Biology and Psychology, University of Göttingen, Göttingen, Germany, **4** Leibniz-Science Campus Primate Cognition, Göttingen, Germany, **5** Bernstein Center for Computational Neuroscience, Göttingen, Germany

\* psaxena@dpz.eu

This is a Registered Report and may have an associated publication; please check the article page on the journal site for any related articles.

## Abstract

Human visual perceptual performance is strongly dependent on a given stimulus' *distance* from the line of sight, i.e. its eccentricity. In addition, multiple studies have shown a dependence on a stimulus' *angular* position relative to the fovea. In humans, the resulting spatial profile of perceptual performance (the "performance field") typically shows better performance near the lower vertical meridian, compared to the upper vertical meridian, and better performance near the horizontal meridian compared to the vertical meridian. Predominantly, these variations have been interpreted as sensory inhomogeneities. But it has also been shown that they are modulated by the allocation of spatial attention, either homogeneously elevating performance or compensating for the sensory inhomogeneities. Here, we propose a study protocol for pre-registration to investigate such interactions between sensory and attentional effects. First, we will determine performance fields for time-dependent, *dynamic* stimuli, namely the direction discrimination of moving random dot patterns. Then, we will establish whether directing focal attention to a particular stimulus location *differentially* improves thresholds compared to a distributed attention condition.

## Introduction

The human visual field is characterized by a central area (the 'fovea') of very high spatial resolution and a steep drop-off towards the much larger peripheral visual locations. Nevertheless, vision outside the fovea provides essential information for assessing the overall visual environment and for identifying potentially relevant visual aspects that warrant further detailed inspection by foveation.

In addition, it has been clear for over two decades that peripheral vision is not only characterized by the loss of resolution with increasing eccentricity, but also by variations at iso-eccentric locations, depending on their angular position relative to the fovea. Such a spatial profile

**Data Availability Statement:** All relevant data from this study will be made available upon study completion.

**Funding:** This project is supported by grants to ST from the Deutsche Forschungsgemeinschaft (DFG, https://www.dfg.de/en): Research Unit 1847-A1, project number 211740722 and Collaborative Research Center 889-C04, project number 154113120. The funders had and will not have a role in study design, data collection and analysis, decision to publish, or preparation of the manuscript.

**Competing interests:** The authors have declared that no competing interests exist.

of performance measures, termed "performance field" has been reported for a wide range of visual tasks [1–10]. Typically, performance is higher in the lower visual hemifield compared to the upper visual hemifield (vertical meridian asymmetry-VMA), and along the horizontal than the vertical axis (horizontal-vertical anisotropy-HVA).

These sensory inhomogeneities have been suggested to arise, to some extent, from retinal factors [11], for example, an inhomogeneity in the retinal distribution of the midget ganglion cells [12]. A contribution from cortical elements has also been suggested, such as the smaller population receptive fields and a correspondingly larger cortical magnification factor along the horizontal compared to the vertical meridian in human early visual areas [13–15].

It has been suggested that attention plays an important role in the visual system's approach to such inhomogeneities. Attentional effects on performance, sensitivity and thresholds might not be identical across the visual field, in effect compensating for sensory inhomogeneities. Evidence for such effects comes from studies that show attentional inhomogeneities [for appearance [3], for speed of information accrual [5], for aspects of motion perception in two quadrants [6], and for processing objects distributed in depth [16].

It should be noted that these attentional inhomogeneities have typically been investigated using static stimuli, in effect focusing on cortical processing along the temporal pathway. This leaves open the possibility that attentional inhomogeneities are weak or absent for stimuli processed along the dorsal pathway, because the sensory inhomogeneities might be weak for such stimuli [17].

Therefore, we plan to focus on dynamic stimuli, designed for dorsal pathway processing by 1) documenting any sensory inhomogeneities in the human performance field for visual motion stimuli, using direction discrimination thresholds as the performance metric and 2) by establishing whether directing the focus of spatial attention to various iso-eccentric stimulus locations *differentially* improves thresholds, depending on their respective angular position relative to the fovea.

## Materials and methods

### Participants

We will recruit participants for our study from a subject pool at the Göttingen Campus. For the power analysis, we simulated data from 20 participants. On each data point, 1000 fits of linear mixed effect model ('lme4' in R [18]) were computed. Results indicate that the simulated dataset allow the detection of a combined effect of attention, location, and cardinal directions with 89% power. Therefore, we will collect data from 10 males and 10 females (age range: 18–35 years). They will have either normal or corrected to normal vision. If an individual participant's data is excluded from the study, for example, due to issues relating to the participant's performance or if a subject did not complete all trials, we will recruit substitute participants. Data from the substituted participants will be made available with the reason of their removal in the appendix. All participants will have to give an informed written consent prior to participating in the study. This study has been approved by the Ethics Committee of the Georg-Elias-Mueller-Institute of Psychology, University of Göttingen (GEMI 17–06–06 171), and is in accordance with the principles of the Declaration of Helsinki.

### Experimental setup

The experiment will be conducted in a dimly lit room with stimuli presented on an LCD monitor (Asus, ROG XG27AQ) with a refresh rate of 120 Hz and a luminance of 0.1 cd/m$^2$. The experiment will be controlled by the Mworks software v0.90 (mworks.github.io). Participants will provide responses using a standard keyboard (Apple). Participants' heads will be stabilized

with a chin-and-forehead rest that is positioned 57 cm in front of the monitor. Participants will be required to fixate on a small white cross (arm lengths: 0.3 degrees, luminance: 84 cd/m$^2$) presented at the center of the screen throughout each trial. We will monitor the participants' eye movements using a binocular Eyelink 1000 system (SR-Research, Ottawa, ON, Canada) at a sample rate of 500 Hz.

## Stimuli

Motion direction thresholds will be measured using moving random dot patterns, consisting of 200 bright dots (speed: 3 degrees/second; density: 5 dots/degrees$^2$; diameter: 0.2 degrees; luminance: 84 cd/m$^2$) randomly positioned within a circular aperture (diameter: 3.2 degrees). In a given trial, four such stimuli will be placed on the two cardinal axes of the screen, centered at an eccentricity of 6.4 degrees (Up, Down, Left, and Right) (Fig 1A).

The dots will move linearly within the stationary aperture along a direction angled clockwise or counterclockwise from one of the four cardinal directions transecting the stimulus (Fig 1C).

## Paradigm / trial structure

Each trial will begin only when a participant directs and maintains his/her gaze at the central fixation cross and pressing the starting key (Fig 2).

In a "focused attention" condition, we will present a short line (the pre-cue; length: 0.34 degrees; line thickness: 0.14 degrees; eccentricity: 0.58–0.92 degrees), pointing towards one of the four stimulus locations (Fig 1B).

In a "distributed attention" condition, all four pre-cues will be presented simultaneously (Fig 1B). Irrespective of the attention condition, after the stimulus presentation, a 100% valid post-cue will indicate the location of the target stimulus. In a block, the two attention conditions will be presented interleaved.

In each block of trials all four RDPs will contain a motion direction close to the same cardinal direction (Fig 1A). Each participant will perform two blocks of trials, differing in the cardinal direction used for the four RDPs (picked randomly).

In a given trial, the motion direction of a target stimulus will be determined using two interleaved 2-up-1-down staircases. The two randomly interleaved staircases will begin ± 10 degrees from the cardinal direction used within that block of trials. In a 2-up-1-down staircase, the direction in a given trial associated with that staircase (the staircase value) will be closer to the cardinal direction if the participant's response was correct in the two previous trials for that target location and associated with that staircase. After each incorrect response to a target at a given location and a given staircase the next direction of a target at that location and associated to the given staircase will be further from the cardinal direction used in this block of trials.

The motion directions of each of the three distractors will be the picked randomly from the two staircase values for that location.

In the focused attention condition, only one pre-cue will appear for 400ms indicating the location of the upcoming target stimulus. In the distributed attention condition, all four pre-cues will be presented, leaving the participant uncertain about the target stimulus location. Following the pre-cue, and an inter stimulus interval (ISI) of 67ms, four moving RDPs will be presented for 300 ms (41 screen refreshes). Following the RDP presentation and an inter-stimulus interval of 45 ms a 100% valid response-cue will appear for 667 ms. This response-cue will allow us to dissociate the effect of attention on the performance from the effect of target uncertainty, especially in the distributed attention condition. Participants will then need to report the target stimulus location and target stimulus' direction by pressing one of the eight keys

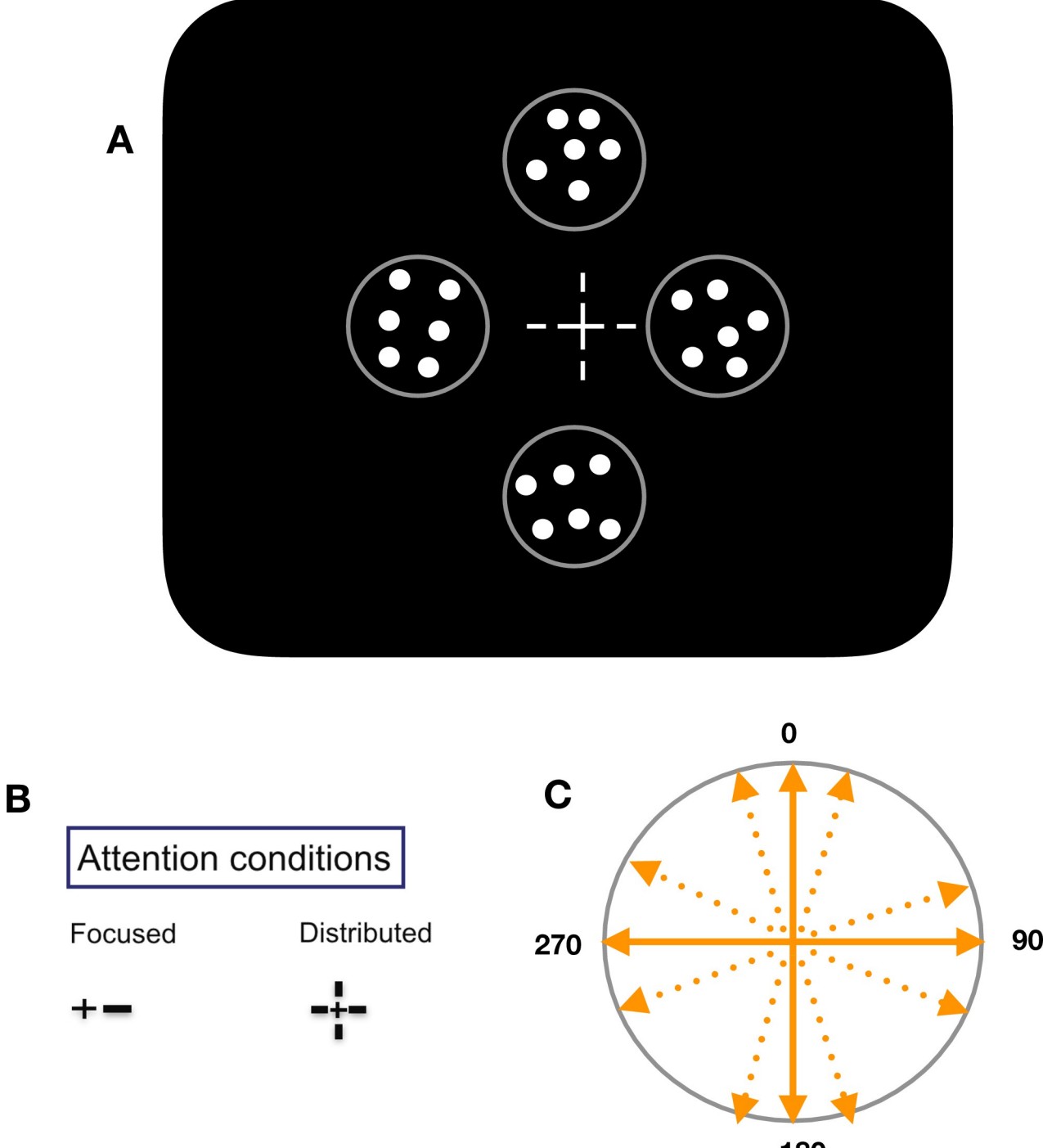

**Fig 1. Stimuli and descriptions.** A) In a given trial, four stimuli will be placed on the two cardinal axes of the screen—up, down, left, and right. B) The two attention conditions: "focused attention" condition—where a short line originating from the fixation cross will point to target stimulus and "distributed attention" condition—where four pre cues will be presented simultaneously, cueing all four stimuli. C) In a given session, all four stimuli will move slightly deviating along one of the four cardinal axes—vertical upward, vertical downward, horizontal rightward, and horizontal leftward.

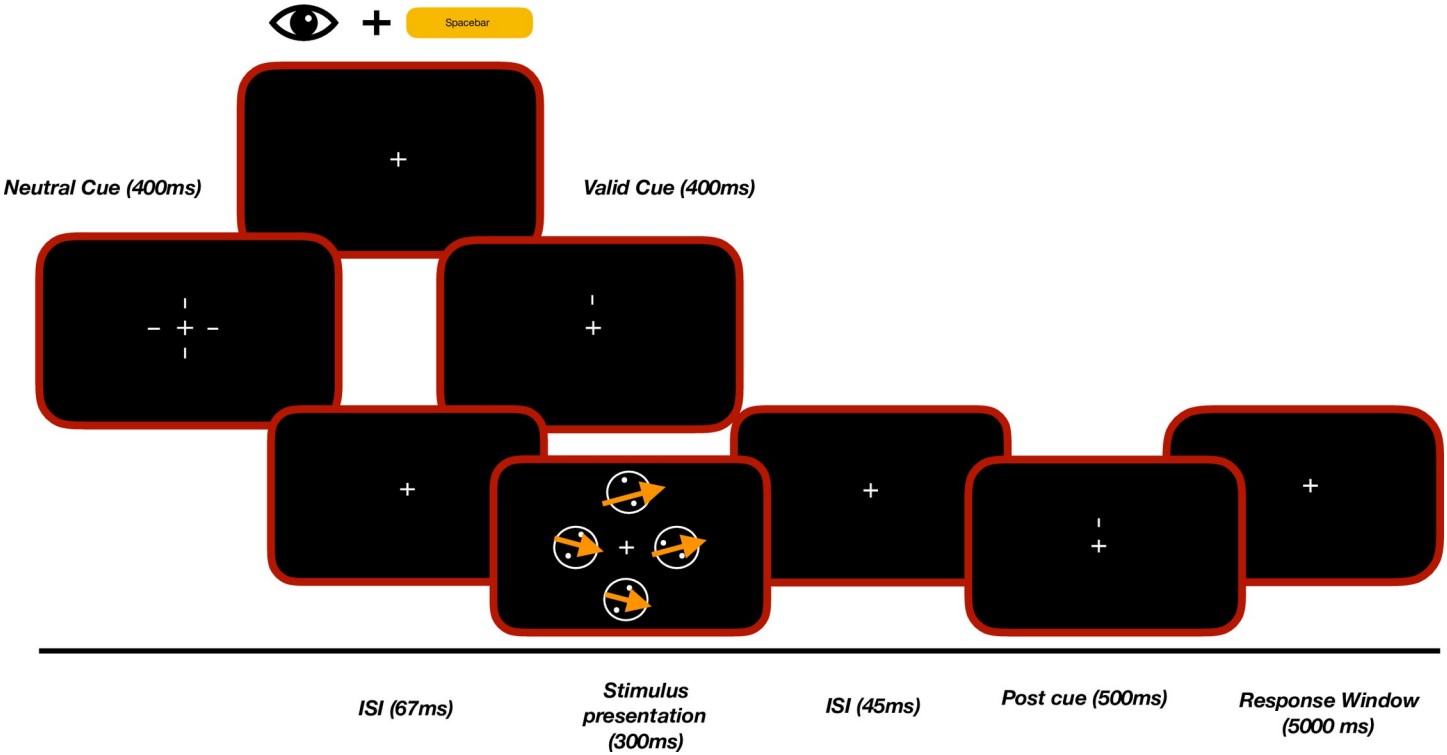

**Fig 2. Schematic flowchart of the experimental procedure.** A trial will begin with participants foveating the fixation cross and pressing the spacebar on the keyboard. Then either four pre cues or one pre cue will appear on the screen to define the trial as either "distributed attention" or "focused attention". After the presentation of the four RDPs, a 100% valid response-cue will indicate the location of the target stimulus. Following which, subjects will respond by the key on the keyboard matching the location of the target stimulus and its motion direction.

(Fig 3B). In case participants do not complete the two responses within 5000 ms, the trial will be tagged as a "no response" trial, and will be queued up again for the subsequent trials.

## Training and experimental sessions

Every subject will go through at least three sessions on separate days (Fig 4). Each session consists of blocks of trials. The first sessions will be training sessions. During these, the rate of

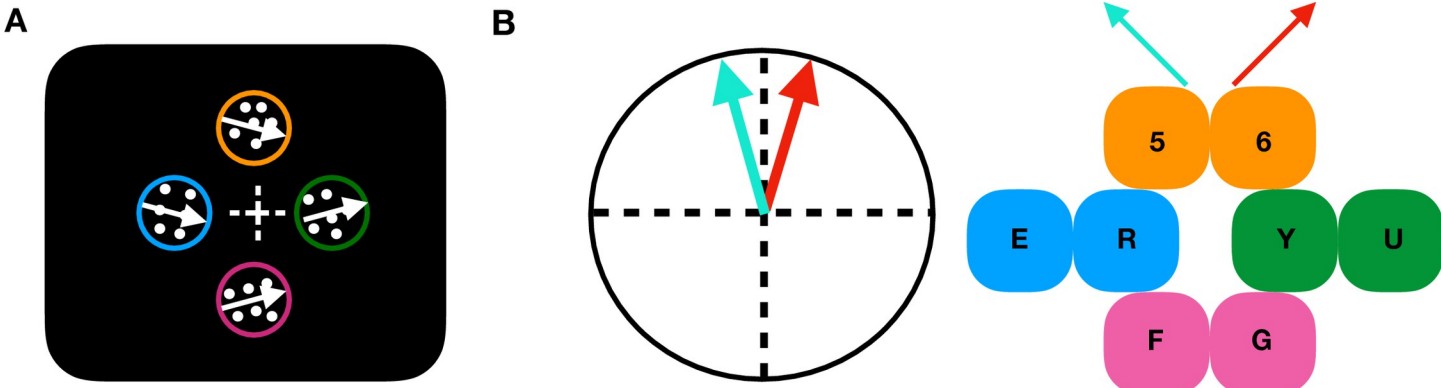

**Fig 3. Locations and responses.** A) Example trial, showing the four RDPs moving along one cardinal direction (here, rightward). Similarly, in another block, the four RDPs can move along one of the three other cardinal directions. B) Response keys used for a block of trials with the upward cardinal direction. Subjects indicate both, the target stimulus' location and the target stimulus' direction, by pressing one of the eight keys. Every participant will be tested on two blocks (differing in the cardinal direction used and the eight response keys).

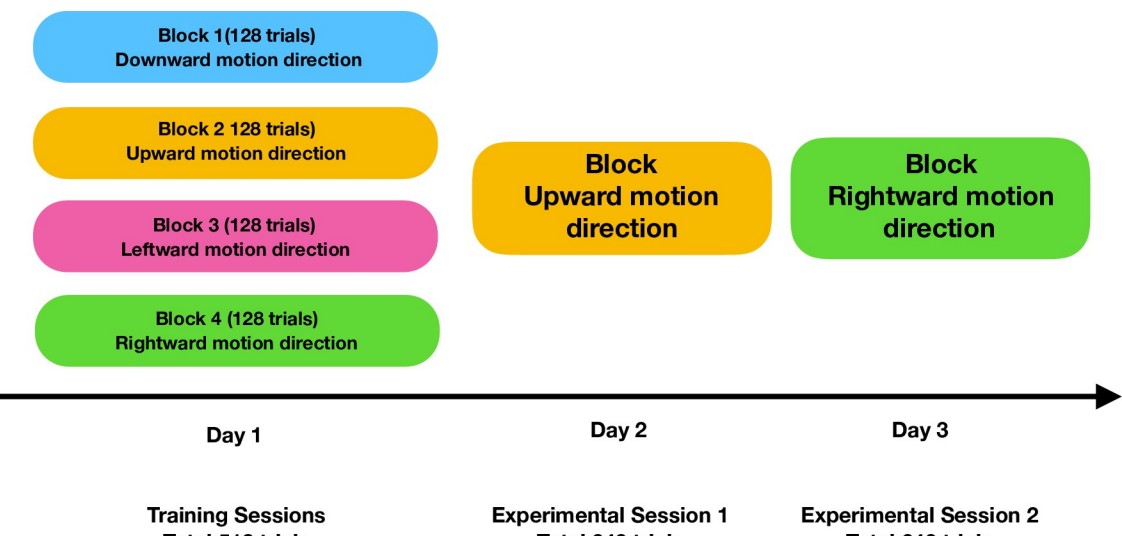

**Fig 4. Schematic diagram of the training and experimental sessions.** Each participant will go through 4 blocks of training blocks on day 1. On the subsequent two days, participants will go through two experimental sessions with one block of 640 trials. The two experimental sessions are defined by the one of the four cardinal directions along which the stimuli are moving–upward, downward, leftward, and rightward. For example, here, day 2 will have one block of 640 trials where motion direction of all four stimuli is along rightward cardinal direction.

incorrect target responses and the staircases will be monitored online, to adjust the difficulty levels to a subject's training level. In a training session, participants complete four blocks of 128 trials each, one for each cardinal direction.

The last two sessions will be experimental session. The sessions will be held on two consecutive days. Each session will consist of a block of 640 successful trials and hence across the two sessions, data will be collected on 1280 trials per participant. To avoid fatigue, participants are encouraged to take short breaks at any time, by delaying the start of the next trial (either by not foveating the fixation cross or by not pressing the space bar). Each block randomly interleaves target locations and the two attentional conditions, to prevent an influence of residual training effects during the experimental sessions.

## Data analysis

The two major hypotheses to be tested in this study are, 1) The direction discrimination thresholds in each of the four locations vary from each other (H1 and H0 depicted in Fig 5B, left panel). 2) application of attention will result in maximum compensation in the discrimination threshold for location where it is the highest (H1 and H0 depicted in Fig 5B, right panel). These two hypotheses will be tested separately for each of the four cardinal directions. Direction discrimination thresholds will be determined by the slope of a cumulative normal psychometric function, using the difference between the x-value at the performance of 0.50 and 0.84. The method of maximum likelihood criterion will be used to define the best estimate of the direction discrimination threshold (MATLAB Toolbox Palamedes v1.11.5). The fitting has four parameters (alpha, beta, gamma, and lambda). Alpha and beta are parameters of interest as they define the point of subjective equality and the slope and are defined as free parameters. Gamma and lambda are left and right asymptotes of the psychometric function and we fix them at 0 and 1, respectively.

The direction discrimination thresholds will be computed separately for the four stimulus locations (UP, DOWN, LEFT, and RIGHT) as well as for the two attention conditions (FOCUSED and DISTRIBUTED).

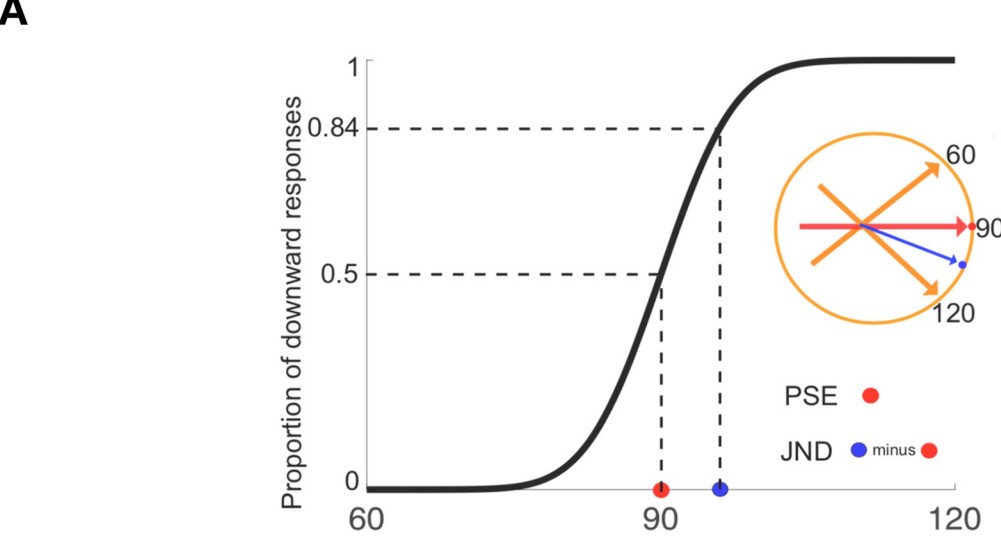

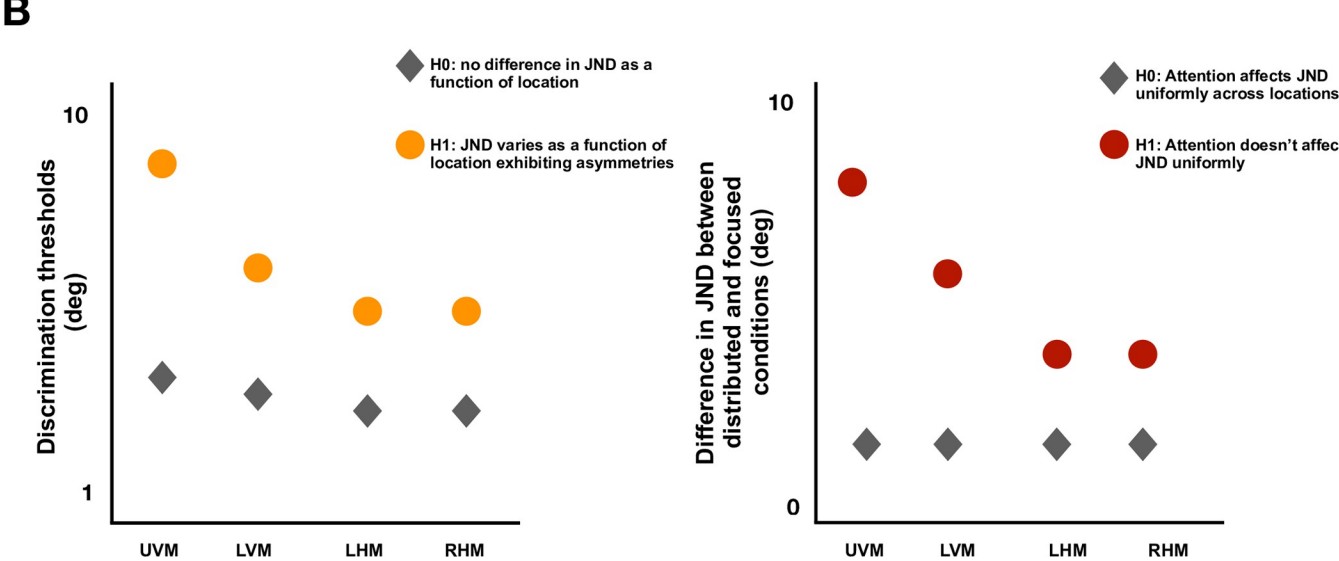

**Fig 5.** A) An example of the cumulative normal psychometric function. A psychometric function characterizes a participants' performance along some varying aspect of the stimulus–here, its motion direction relative to a reference direction. B) This experiment tries to establish whether cognitive facilities like attention interact with the sensory encoding, affecting the perception at visual locations with poorer perceptual performance to compensate for this deficit. Depicted here are schematic plots from testing effect 1 and effect 3 (items from the list above).

In order to determine whether the direction discrimination thresholds were influenced by attention, location, and cardinal direction, we will use a repeated measures analysis of variance (ANOVA) and a linear mixed effect model (LMM). Both will include the abovementioned factors and their interactions up to the third order. In the LMM these factors will be included as fixed effects. Additionally, to account for potential day-to-day variation in the individual discrimination thresholds, we will include a fixed effect of the test day to the random intercept

effect of subject ID. Furthermore, we will include all theoretically identifiable random slopes, as their inclusion has been shown to be important to avoid an overconfident model and keep type I error rates below nominal 0.05 [19, 20].

In summary, we will determine the strength of the following four effects/interactions:

1. A direct effect between RDP-locations and discrimination thresholds (JND).

2. A 2-way interaction of motion directions X RDP-locations on JND.

3. A 2-way interaction of attention conditions X RDP-locations on JND

4. A 3-way interaction of attention X motion directions X RDP-locations on JND.

## Author Contributions

**Conceptualization:** Pankhuri Saxena, Stefan Treue.

**Data curation:** Pankhuri Saxena.

**Funding acquisition:** Stefan Treue.

**Investigation:** Pankhuri Saxena.

**Methodology:** Pankhuri Saxena, Stefan Treue.

**Project administration:** Stefan Treue.

**Resources:** Stefan Treue.

**Software:** Pankhuri Saxena, Stefan Treue.

**Supervision:** Stefan Treue.

**Writing – original draft:** Pankhuri Saxena.

**Writing – review & editing:** Pankhuri Saxena, Stefan Treue.

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
