## [Decision Letter · Decision Letter 0]

14 Apr 2023

PONE-D-23-01491Effect of attention on motion processing in humans around iso-eccentric location in the visual field: a registered report protocolPLOS ONE

Dear Dr. Saxena,

Thank you for submitting your manuscript to PLOS ONE. After careful consideration, we feel that it has merit but does not fully meet PLOS ONE’s publication criteria as it currently stands. Therefore, we invite you to submit a revised version of the manuscript that addresses the points raised during the review process.

The two reviewers indicate some minor issues, concerning the methodology used and some bibliographic additions. Note that reviewer #1 indicates that there is no accessible data, and that it is PLOS editorial policy to make all data accessible. I therefore ask the authors to respond to this aspect, also considering the type of article proposed.

Also note that the work of reviewer #2 is contained in the attachment.

We look forward to receiving your revised manuscript.

Kind regards,

Nicola Megna, M.D.

Academic Editor

PLOS ONE

Journal Requirements:

2. In your cover letter, please confirm that the research you have described in your manuscript, including participant recruitment, data collection, modification, or processing, has not started and will not start until after your paper has been accepted to the journal (assuming data need to be collected or participants recruited specifically for your study). In order to proceed with your submission, you must provide confirmation.

3. We note that Figures 2 an 3 in your submission contain copyrighted images. All PLOS content is published under the Creative Commons Attribution License (CC BY 4.0), which means that the manuscript, images, and Supporting Information files will be freely available online, and any third party is permitted to access, download, copy, distribute, and use these materials in any way, even commercially, with proper attribution. For more information, see our copyright guidelines: http://journals.plos.org/plosone/s/licenses-and-copyright.

a. You may seek permission from the original copyright holder of Figures 2 and 3 to publish the content specifically under the CC BY 4.0 license. 

Reviewers' comments:

Reviewer's Responses to Questions

**Comments to the Author**

1. Does the manuscript provide a valid rationale for the proposed study, with clearly identified and justified research questions?

Reviewer #1: Yes

Reviewer #2: Yes

2. Is the protocol technically sound and planned in a manner that will lead to a meaningful outcome and allow testing the stated hypotheses?

Reviewer #1: Yes

Reviewer #2: Yes

3. Is the methodology feasible and described in sufficient detail to allow the work to be replicable?

Reviewer #1: Yes

Reviewer #2: Yes

4. Have the authors described where all data underlying the findings will be made available when the study is complete?

Reviewer #1: No

Reviewer #2: Yes

5. Is the manuscript presented in an intelligible fashion and written in standard English?

Reviewer #1: Yes

Reviewer #2: Yes

6. Review Comments to the Author

You may also provide optional suggestions and comments to authors that they might find helpful in planning their study.

Reviewer #1: Dear authors,

I read with interest the manuscript PONE-D-23-01491 entitled “Effect of attention on motion processing in humans around iso-eccentric location in the visual field: a registered report protocol”.Specifically, the authors plan to examine how the sensory asymmetries found in peripheral vision interact with the attentional effects. The introduction is short, but well written and explains the theoretical background. The methods section reports all relevant information, but only requires a few modifications. I report my suggestions below.

1) In the general field of open science, you should provide detailed information to facilitate replicability of the study. It seems counter-intuitive for science, but recent papers suggest that replicability of methods is lacking. See Asendorpf et al., 2016 for details.

2) In the participant section, please insert a power analysis in order to demonstrate that the sample size is adequate. If you use a mixed linear model for the analysis (see below), please also use this approach in this part (I suggest using the simr package in the R statistical environment).

(3) Include in the description of participants the "final sample size", but for transparency, all discarded participants should be reported with the reason for their removal. Take into account that removing some participants is normal (participants who did not complete all sessions, inaccurate measurements, etc.), but more importantly, all data removed needs to be reported for transparency, along with the criteria used.

4) I recommend inserting a schematic representation of the overall procedure (session and block) after figure 3.

5) In the analysis section I suggest using the Linear Mixed Model (LMM) instead of GLM. I suggest using ID as a random factor.

Asendorpf, J. B., Conner, M., De Fruyt, F., De Houwer, J., Denissen, J. J., Fiedler, K., ... & Wicherts, J. M. (2016). Recommendations for increasing replicability in psychology.

Reviewer #2: Please see attached file.

7. PLOS authors have the option to publish the peer review history of their article (what does this mean?). If published, this will include your full peer review and any attached files.

Reviewer #1: No

Reviewer #2: **Yes: **Marisa Carrasco

---

## [Author Response · Author response to Decision Letter 0]

7 Jun 2023

See attached 'Response to Reviewers' file.

---

## [Decision Letter · Decision Letter 1]

19 Jul 2023

Effect of attention on human direction-discrimination thresholds at iso-eccentric locations in the visual field: a registered report protocol

PONE-D-23-01491R1

Dear Dr. Saxena,

We’re pleased to inform you that your manuscript has been judged scientifically suitable for publication and will be formally accepted for publication once it meets all outstanding technical requirements.

Kind regards,

Nicola Megna, M.D.

Academic Editor

PLOS ONE

Additional Editor Comments (optional):

Reviewers' comments:

Reviewer's Responses to Questions

**Comments to the Author**

1. Does the manuscript provide a valid rationale for the proposed study, with clearly identified and justified research questions?

Reviewer #1: Yes

Reviewer #2: Yes

2. Is the protocol technically sound and planned in a manner that will lead to a meaningful outcome and allow testing the stated hypotheses?

Reviewer #1: Yes

Reviewer #2: Yes

3. Is the methodology feasible and described in sufficient detail to allow the work to be replicable?

Reviewer #1: Yes

Reviewer #2: Yes

4. Have the authors described where all data underlying the findings will be made available when the study is complete?

Reviewer #1: No

Reviewer #2: Yes

5. Is the manuscript presented in an intelligible fashion and written in standard English?

Reviewer #1: Yes

Reviewer #2: Yes

6. Review Comments to the Author

You may also provide optional suggestions and comments to authors that they might find helpful in planning their study.

Reviewer #1: All modification and suggestions were performed and consequently the manuscript could pass to the next stage.

Reviewer #2: This revised version is much better and I look forward to the results of the study.

Minor comments:

(1) Ref 17 appears before 14-16

(2) Ref16 should be Himmelberg, Winawer & Carrasco Nature Communications, instead of TINS.

Although the TINS review is also relevant for this study and could be cited elsewhere.

(3) Make sure all references that appear in the reference list are cited in the manuscript.

7. PLOS authors have the option to publish the peer review history of their article (what does this mean?). If published, this will include your full peer review and any attached files.

Reviewer #1: No

Reviewer #2: **Yes: **Marisa Carrasco

---

## [Editor Report · Acceptance letter]

25 Jul 2023

PONE-D-23-01491R1 

Effect of attention on human direction-discrimination thresholds at iso-eccentric locations in the visual field: a registered report protocol 

Dear Dr. Saxena:

I'm pleased to inform you that your manuscript has been deemed suitable for publication in PLOS ONE. Congratulations! Your manuscript is now with our production department. 

Kind regards, 

on behalf of

Dr. Nicola Megna 

Academic Editor

PLOS ONE